# Description of the Distribution Law and Non-Linear Dynamics of Growth of Comments Number in News and Blogs Based on the Fokker-Planck Equation

**Dmitry Zhukov [1], Julia Perova [2,\*] and Vladimir Kalinin [1]**

[1] Institute of Cybersecurity and Digital Technologies, MIREA-Russian Technological University, 78 Vernadsky Avenue, 119454 Moscow, Russia; zhukovdm@yandex.ru (D.Z.); vkalininz@mail.ru (V.K.)

[2] Institute of Radio Electronics and Computer Science, MIREA-Russian Technological University, 78 Vernadsky Avenue, 119454 Moscow, Russia

[\*] Correspondence: jul-np@yandex.ru; Tel.: +7-916-368-05-34

**Abstract:** The article considers stationary and dynamic distributions of news by the number of comments. The processing of the observed data showed that static distribution of news by the number of comments relating to that news obeys a power law, and the dynamic distribution (the change in number of comments over time) in some cases has an S-shaped character, and in some cases a more complex two-stage character. This depends on the time interval between the appearance of a comment at the first level and a comment attached to that comment. The power law for the stationary probability density of news distribution by the number of comments can be obtained from the solution of the stationary Fokker-Planck equation, if a number of assumptions are made in its derivation. In particular, we assume that the drift coefficient $\mu(x)$ responsible in the Fokker-Planck equation for a purposeful change in the state of system $x$ ($x$ is the current number of comments on that piece of news) linearly depends on the state $x$, and the diffusion coefficient $D(x)$ responsible for a random change depends quadratically on $x$. The solution of the unsteady Fokker-Planck differential equation with these assumptions made it possible to obtain an analytical equation for the probability density of transitions between the states of the system per unit of time, which is in good agreement with the observed data, considering the effect of the delay time between the appearance of the first-level comment and the comment on that comment.

**Keywords:** nonlinear dynamics; processes in social systems; Fokker-Planck equation; power law; monitoring; management

## 1. Introduction

The description of social network behavior and information resources is one of the most important areas of mathematical sociology. From a practical point of view, the development of models describing user opinion dynamics and preferences contributes to the development of systems for automated monitoring of the public mood and its changes. Compared to traditional methods of studying public opinion, the advantage of such systems is that of automated information processing. Social surveys require the development of questionnaires and sampling, which is complicated by the necessity to cover all strata of society. In addition, respondents tend to provide socially desirable responses.

Another advantage of automated information processing for social networks and comments to newsfeed is that it identifies straightforward comments related to a socially significant topic and to highly-publicized news. Therefore, the development of automated information processing tools provides feedback between society and government bodies, starting from the municipal level and ending at the level of state authorities.

The development of automated tools assumes that their work should be based on algorithms based on approved mathematical models. In addition, it is of the utmost

importance not only to monitor and analyze the processes involved in research but also to predict their evolution, which is necessary to ensure sustainable social development.

The dynamics of the changes in opinions and moods of Internet users can be largely attributed to stochastic processes, but with the possibility of targeted impact. On the one hand, the human factor (many people with different opinions, preferences, and behavior patterns) creates random changes (due to the wide variety of behavioral models of users). On the other hand, elements of opinion consistency are introduced into the dynamics of changes. A detailed description of the use of stochastic methods for modeling the dynamics of social processes can be found in [1].

In this regard, we consider models based on the Fokker-Planck equation to be the most promising to develop models of the changes in public mood dynamics, which takes into account both ordered and random changes.

The Fokker-Planck equation is widely used for analyzing and modeling the behavior of time series when describing processes in complex systems [2–5], for example, when analyzing the dynamics of the non-stationary time series of stock and commodity indices. To predict changes, based on the Fokker-Planck equation and sample data, the distribution functions of the series levels are constructed in the form of a sum of polynomials in which the coefficients of drift and diffusion may depend on a specific parameter, which is the level of the series according to various laws and is empirical.

It should be noted that, apart from the Fokker-Planck equation, other approaches are used for modeling based on differential equations, for example, the Liouville equations [5,6], the diffusion equations [4,7] and many others. A detailed review of modeling social processes is presented in [8].

The Fokker-Planck equation is a second-order partial differential equation that not only contains a term responsible for stochastic changes ("diffusion"), but also an element responsible for opinion consistency ("draft"). From the Fokker-Planck equation, it is possible to obtain a probability density function of transitions per unit of time between states of a system. A system can be defined as a blog or newsfeed that users comment on, and its state will be the number of comments that are observed at a given time.

In addition to describing dynamic processes, stationary solutions can be obtained from the Fokker-Planck equation, which can describe the state of a system in a stationary state, when, for example, its evolution has already ended, and changes do not occur. One example of such stationarity may be the final static distribution of newsfeed or blogs by the number of comments on them.

The study of processes occurring in complex systems with the participation of the human factor shows that very often a power law of distribution is performed for the observed characteristics of the parameters of these processes. If we imagine the interconnection of the elements forming a complex system as a diagram, it turns out that the networks that arise in this case—social, communication, Internet link networks, citations, and others—are well described by scale–free models (scale-invariant), in which the degrees of vertices (nodes) are distributed according to the power law $p(x) \sim x - \gamma$ (where $\gamma$ is the characteristic degree) [9–14]. Scale-free networks are self-similar, i.e., in any part of the network, the distribution of degrees will be the same.

The power model is widely used in the analysis of processes in complex social systems, but at the same time the issue of the theoretical justification of the possibility for its application requires further study. In our opinion, this justification is very crucial. The identification of the nature of the processes from which the power law arises is necessary for a deeper study of behavior and analysis of complex social systems.

In addition, we are not aware of attempts to apply a theoretical description of processes in social networks and network mass media based on the Fokker-Planck equation from the standpoint of formulating and solving boundary value problems based on it.

The purpose of our work is to investigate the possibility of obtaining from the Fokker-Planck equation, often observed in practice in complex social systems, the power law of

the distribution of parameters of the processes occurring, and to show that under certain assumptions this equation can be used to describe both static and dynamic characteristics.

## 2. Research Methods

As described in the lead section, our article is devoted to solving the following issues. First, we collected statistics on the dynamics of changes for the number of comments on the news on the feed portal of the Russian radio station «Echo of Moscow» https://echo.msk.ru/ (accessed on 13 September 2021) (one of the leading Russian commercial radio stations and newsfeeds). Then, we describe the processing of the collected data and the results obtained (in particular, in a stationary state, a power law of the news distribution by the number of observed comments). The observed dynamics of change in the number pf comments (to news feeds and blogs) is described by either two-stage or S-shaped curves. Further, using the stationary Fokker-Planck equation and a number of assumptions about the dependence of coefficients describing a random and purposeful change in the state of the system $x$ ($x$ is the current number of comments on the news) on the magnitude of this current state $x$, we derive from the Fokker-Planck equation the power law of the distribution of news by the number of comments. Then, based on the Fokker-Planck equation, we construct a dynamic model of changes in the state of the systems under consideration over time. The analysis of the models showed good agreement with the observed characteristics of the processes. This suggests that the models we have developed can be used not only to analyze social processes, but also to predict their evolution, which is very important for managing the stable development of social relations. In conclusion, we discuss the possible application of our models in practice and the creation of algorithms for automated systems for the monitoring of public opinion.

## 3. A Brief Overview of Existing Studies of the Structure of Complex Social Systems and the Processes Observed

One of the directions in the study of complex networks is the study of their structure, based on the possibility of representing processes at the graph level, using a set of attachments at the level of individual nodes for data aggregation (of the properties of the whole from the properties of the quotient). Aggregation is crucial, since it should, in principle, provide an isomorphism-invariant representation of the graph, i.e., the representation of the graph should be a function of the nodes of the graph, considered as some set.

In [15], the DeepSets aggregation operator based on self-organizing maps (SOM) is considered. Using SOM allows calculation of representations of nodes that include information about their resemblance. Experimental results on real data sets show that the proposed approach provides improved predictive performance compared to the generally accepted summing aggregation and many modern graph neural network architectures in the literature.

Since, with the growth of the network, the search for similarities between nodes in the network is a time-consuming process, to optimize the process of solving problems of predicting connections and detecting communities' researchers in [16] use swarm algorithms. Swarm-based optimization methods used in social network analysis are compared in this article with community analysis and link analysis based on traditionally used approaches.

In [17], the authors consider the mathematical model of mixed membership in user groups, which are formed stochastically. This preliminary solution the authors base on the method of detecting pairwise measurements, which subsequently show the presence or absence of connections between a pair of nodes. When analyzing the approach for probabilistic changes between pairs of objects, it is usually necessary to introduce assumptions, for example, independence, or assumptions of the inconsistency of this connection (mixed membership in stochastically forming groups). The proposed model allows, under certain assumptions, the tracking of dynamic changes in the number of nodes in the forming of groups and their clustering by groups.

In the presented model, from the development of choice and influence on social networks [18], the authors consider a model for which the number of nodes and the network topology (structure of connections) are dynamic. A significant disadvantage of this model is that it explicitly considers the connections between all pairs of nodes. This action leads to quadratic difficulty in calculating the change in the number of participants in various social groups and a significant increase in the calculation time. It is worth noting that real social networks and systems are sparse. This means that most participants do not have paired connections, and the number of their connections is itself random. Introducing the concept of sparsity into the model [18], as well as taking into account the random nature of the number of connections for each node (user) of the network, can significantly increase the speed and efficiency of using this model.

The authors in [19] use a structure analysis technique that dynamically develops, and therefore has a multimodality of, the graph of the social network. Using this approach to real graph structures in practice shows that there is temporary online regularity in people's social interactions. Moreover, correlations are found between the occurrence of friendship between participants and the settings of the interactive social network. Separately, it is worth noting that physical contacts between people can be considered as an interactive dynamically changing network.

In article [20], the authors described methods of structuring and influencing the dissemination of information on mobile social networks. In these networks, a group of users is typically treated as some kind of entity in which individuals can exchange messages. The authors also note that there is a variety of models for analyzing the dissemination of information on mobile social networks, but none of the existing methods considers the concept of information dissemination in the group. Therefore, the authors of the paper used the SIR model, which is used to spread viruses in computer networks, and applied this to the dynamics of the information dissemination process in groups. Simulations using the Monte Carlo method showed that group propagation increases the overall speed of information propagation on the network. In addition, the authors note that the presence of groups with a significant number of participants is most effective in disseminating information than the presence of a huge number of groups but with a small number of participants. This analysis of the impact of the structure on the dissemination of information within it proves that their distribution in the networks of Erdesh-Rennie and Barabashi-Albert does not show any differences. Ref. [20] analyses the stochastic model of opinion dynamics in social networks. This model is based on a multi-agent approach, for which the opinion of each network member is randomly influenced by the actions of others (its neighboring nodes). Examples were given that, since the number of users (nodes) in the network is not infinite, the model as a result asymptotically creates consensus. The consensus value usually corresponds to one of the absorbing states of the Markov system. However, when the number of nodes is large, some metastable transition states are observed in places. The duration of these transient states may be as long as desired in time, and the state data may be characterized using the mean field approximation for the Markov system. Ultimately, the authors propose a model by which opinion control in the social network is possible.

We can consider several statistical studies [21–23] that have widely used the method of studying profiles in social networks. The purpose of these studies is to identify the social mobility of people based on their publications accompanied by geodata. The authors found a large number of such publications, and based on these an approximate map of the user's movements was compiled, the main centers of activity were identified, and the person's place of residence was established. According to the data on the place of residence, the people's names were found. Further, using a database of names distributed by gender, it was possible to determine the gender of more than half of all the accounts studied; according to the surname data, the researchers tried to establish information about the race and age of users, successfully in 38% and 14% of cases, respectively. These studies have shown that it is possible to establish some demographic characteristics, knowing only about the movements of a person or knowing his first and last name.

Using the comparison of time slices, it is possible to determine dynamically changing temporary communities of users of social network structures. The study of these dynamic communities makes it possible to significantly simplify the analysis of the dynamics of a complex system of social interactions as it evolves over time.

Consider [24], which presents the fundamental structures of dynamic social networks based on a high-resolution dataset describing a tightly connected population of 1000 first-year students at a large European university. The authors of this article consider the physically short interactions that they measured using Bluetooth, supplemented with information received from telecommunications networks (information about calls and messages), social networks and the demographic and geolocation data of users.

Human social communities by their nature overlap due to individuals participating in several different groups (in the theory of complex networks, such nodes are called jumpers). During the week, meetings of the subjects of the created compact structure take place, either a meeting of friends outside the university, or of all students (such structures are called cores). In a network of short physical interactions, all participants are present at the same time and are in physical contact.

The location of the core members can also be forecast. The objects that helps to do this are the kernels themselves. By observing the usual routes of the people who make up the core and their behavioral habits, it is possible to predict the geographical location of a person in the next time interval with high accuracy (on average in 93% of cases), such high accuracy proving that human mobility patterns are regular. It is also worth noting that the members of the core have fewer location states than individuals, which leads to lower values of information entropy on average.

The condition that geospatial studies are conducted for a part of the social group, yet the study is limited to certain time frames, shows any complex interaction between time, place and social context. It also supports the hypothesis that often. when people are most unpredictable in the geospatial domain, they exhibit some predictable social behavior. Linking the results of this article with the literature on dynamic community detection, it can be noted that there are many methods in the literature that would allow the detection gatherings in everyday life, but here the authors used a simple comparison of graph components to emphasize the fact that emerging social structures are natural, and these complex methods are not needed to determine their occurrence.

In fact, Ref. [24] provides a quantitative assessment of long-term patterns encoded in the micro dynamics for a huge system of interacting nodes, which are characterized by predictability and a high degree of order.

Let us consider another paper on dynamic models [25]. Recent developments in the field of social networks have shifted the focus from static representations to dynamic ones, requiring new methods of analysis and modeling. Observations in real social systems have revealed two main facts that play a very important role in the evolution of networks and affect the current processes of distribution: the strategies that individuals adopt when choosing between new or old social systems, connections, and the turbulent nature of social activity that sets the pace of these choices. The results are verified using numerical simulation and compared with two observable data sets.

In [26], methods of assessing public opinion and highlighting the mood of users are carried out using a method based on the use of vocabulary and semantics and inherited from the classical approach to the analysis of public sentiment. Neural networks are used for this method. The task of the neural network is to determine important keywords, which are then checked by experts in this subject area. Formally, the program first analyzes articles and determines how often different words are found in them. Next, the program identifies the most commonly used words and expressions, and makes them significant. Then, on their basis, the program builds a lexicon that characterizes the public mood based on the transmitted news articles.

In [27], the authors described the workings of the algorithm for analyzing certain topics from the social network. In addition to collecting information, there are methods for

processing and sorting information. In addition, the time elapsed between publications is measured so that it is subsequently possible to restore the order of publications and obtain a time scale based on these data. Following from the above, the result is a graph that can be used to track the growth and decline in popularity of certain topics discussed on social networks. You can also trace what moods are accompanied by what events in society. In addition, it is possible to determine the period of active discussion for certain topics.

Article [28] describes the method of studying political sentiments in society, based on the analysis of the social network. This method is carried out by searching for special words in the text that are previously entered in the program database. The main task of this system is to track by how much different political parties are preferable to citizens, and which are less significant. In addition, which topics are most resonant and most discussed in society are monitored. Additionally, with the help of the program, it is possible to find out how many people in percentage terms support a certain political party.

The subject of [29] is that of microblogs. The authors of this study used the method of keyword analysis. With the help of such analysis and machine learning, they managed to divide the initial sample into six age groups and identify the topics that participants in each age group most often discuss and on which they most often express their thoughts. Teenagers under 18 most often discuss sports; young people aged 18–25 most often talk about entertainment; people aged 25 to 30 mainly discuss family and business, older people (31–36 years old) are most interested in technology, users aged 26–40 begin to worry about their health and speak about this more often, and those over 40 like to discuss politics. Thus, the most frequent topic for discussion was determined for each age group; this does not mean that each member of this group necessarily discusses this topic, but it is more likely that the person discussing this topic belongs to this age group.

The authors of [30–34] proposed a method that evaluates the mass media according to several criteria (topic, evaluation criteria/properties, classes), which combine thematic modeling of context and multi-criteria decision-making. This evaluation system is based on corporate analysis as follows: the conditional distribution of media probabilities by topic, detail and class is calculated after the formation of the thematic model of corporations. Several approaches, including manual labeling, a multi-corporate approach and an automatic approach, are used to obtain coefficients that show the interaction regarding how each topic relates to each evaluation criterion and to each class described in the document. The multi-corporate approach proposed in the study involves assessing the thematic asymmetry of text enclosures to obtain coefficients describing the relationship of each topic to a certain criterion. These factors, in combination with the thematic model, can be used to evaluate each document in the enclosures according to each of the criteria and classes considered. This method was applied to a body of texts consisting of 804,829 news publications from 40 Kazakh sources, published from 1 January 2018 to 31 December 2019 (over a period of 2 years) to classify negative information on socially significant topics. The study produced a BigARTM model (200 topics) and applied this model, including completion of the analytical hierarchical process table (AHP) and all necessary high-level labeling procedures. The experiments carried out confirm the general possibility of evaluating media using the thematic model of text enclosures, since the classification problem achieved an area estimate under the receiver performance curve (ROC AUC) of 0.81, which is comparable to the results obtained for the same task using the BERT model.

The developed system, in which the proposed model was integrated, allows the solution of classic problems, such as simple reports or sentiment analysis. Moreover, it has a number of unique possibilities for use. It provides options such as automatically analyzing a specific topic, event, or object without having to create a keyword-based query. The analysis is based on an arbitrary list of criteria and not limited to sentiment alone. This list includes social significance, popularity, manipulation, propaganda content, attitude to a certain country, attitude to a certain area, analysis of the dynamic behavior of topics, predictive analysis at the thematic level, etc.

In [35–37], the KroMFac technique is proposed, which performs community detection using regularized non-negative matrix factorization (NMF) based on the Kronecker graph model. KroMFac combines network analysis and community discovery methods in a single unified structure. This technique connects four areas of research, namely the detection of communities on graphs, of overlapping communities, of communities in incomplete networks with missing edges, and of complete networks.

It is possible to consider several works, close to the subject of our research, on the description of processes in complex social network structures.

Article [38] considers a model describing the spatial and temporal distribution of information in social networks based on a partial differential equation. In this paper, a non-autonomous diffusion logistic model with Dirichlet boundary conditions was created and investigated, which showed that the diffusion of data is strongly influenced by the diffusion coefficient and internal growth rate (the spread of information or rumors can be considered as a kind of virus that does not have a physical form).

Article [39] proposes a mathematical model of information dissemination and a mechanism of evolution of the state of the information node using the theory of thermodynamic molecular thermo-diffusion motion in combination with the model of epidemic infection. Four different network topologies are used for the time-varying online social network (OSN) information dissemination process (regular network, small worlds network, random network, and non-scale network).

When distributing OSN information, the concept of information entropy is used. The process of information dissemination determines the transition of the system from one stable state to another. The transfer function is set by such information parameters as information energy, information temperature and energy entropy. The considered model is based on the relationship between the state of microscopic network nodes and the rules of macroevolutionary evolution. The authors of the article conduct simulation experiments and empirical comparative experiments in networks with different topological structures. The proposed model is trained and evaluated using experimental data collected from the Chinese network Baidu.

The authors of article [40] propose a model for describing the distribution of messages in social networks. This proposal is based on systems described by means of differential equations that show the propagation of various information in a network graph chain. The authors are convinced that this model allows the taking into account of specific mechanisms for transmitting messages. In this model, the vertices of the graph are people who, when a message is received, form their attitude to it. After this, people decide on further transmission of this message over the network, provided that the corresponding interaction potential of the two persons exceeds a certain threshold level.

The authors developed a mathematical method for calculating the timing of the distribution of messages in the corresponding graph chain, which is reduced to solving a number of Cauchy problems for systems of ordinary nonlinear differential equations. Formally, these systems can be simplified, and some equations can be replaced by the Boussinesque or Corteweg de Frieze equations. The presence of soliton solutions for these equations gives us reason to consider social and communicative solitons as an effective tool for modeling the processes of disseminating messages on social networks and studying various influences on their distribution. If certain assumptions are allowed, this model, considered in [33] has some analogies with the spread of viral epidemics.

In conclusion, it should be noted that almost no one has studied models based on the Fokker-Planck equation to describe processes in complex network social systems.

## 4. The Analysis of the Observed Statistics of Comments from Users of Newsfeed Resources and Blogs—Statement of the Research Problem

### 4.1. Data Source Selection and Presentation

Newsfeed and blogs, on which Internet users leave their comments, are one of the most important among network objects, since they can indicate public opinion in real time.

A socially significant topic usually attracts both supporters and opponents, who enter into discussions and leave comments. The more highly-publicized the news or blog, the higher the user activity and the greater the number of commentators (a multi-level structure of comments on comments appears). The analysis of the structure of comments of users of news posts and blogs is one of the most practically significant and relevant scientific tasks, the solution to which ensures sustainable social development.

To study the nature of the observed processes and collect data, we have selected the commercial radio station and newsfeed «Echo of Moscow» https://echo.msk.ru/ (accessed on 13 October 2021). The choice is determined by the following reasons:

1.  News portal (The commercial radio station) is among the top 10 news sites in Russia and in July of 2021 took ninth place for attendance and seventh for user activity, also at the end of July 2021 ranking in the top eight of the cited radio stations and occupying first place through hyperlinks in social media at the end of August 2021 and fourth place according to the citation index in the media.
2.  The portal has various themes (presents news from the political, sporting, economic and scientific arenas, cultural orientation, etc.).
3.  The news portal has been in existence since 1990 and has established itself as a reliable, truthful and publicly available news source, and also publishes blogs of well-known media personalities.
4.  There is practically no pre-moderation of comments (pre-moderation applies only to new users or users who have previously violated the rules of the news portal), but there is post-moderation of discussions (the requirements for comments and prohibitions on their placement can be found at the link: https://echo.msk.ru/moderate.html (accessed on 13 October 2021)). Users can express different opinions (which do not have to coincide with the official position) and their comments are deleted only for violating the rules.

At first, we downloaded the news range we were interested in, using a special software application (parser). The portal distributes news by day, and each individual day can be found at the link (https://echo.msk.ru/news, (accessed on 13 October 2021), where day is the day, month is the month, year is the year). Each news item has a number of parameters such as: news text, unique identification number on the portal, title, web page address (URL), metadata (date and time of publication, number of views and comments), texts of user comments (as well as unique identification number of the comment, unique identification number of the user, date and time of comment, comment hierarchy level, relationship by level of commenting to the parent comment) and available information about authors (unique identification number of the user on the webpage, city, occupation, place of work, name or nickname, registration date, the number of recommendations and user profile views, the total number of comments for the observed period, etc.). On average, the number of news items varied from 160 to 190 per day. While collecting the data, we downloaded information about which of the users commented on other users' reaction to news. Based on the data obtained, a database of the newsfeed archive was created.

Figure 1 shows the correspondence of the share of commentators to the number of comments they wrote (the observed density of the distribution of commentators by their number of comments) for the period from 1 January to 31 December 2020. Similar dependencies can be built for any period (day, week, month, quarter, year). The total number of news items published in 2020 was 65,560, of views 196,609,650, and of comments 564,764.

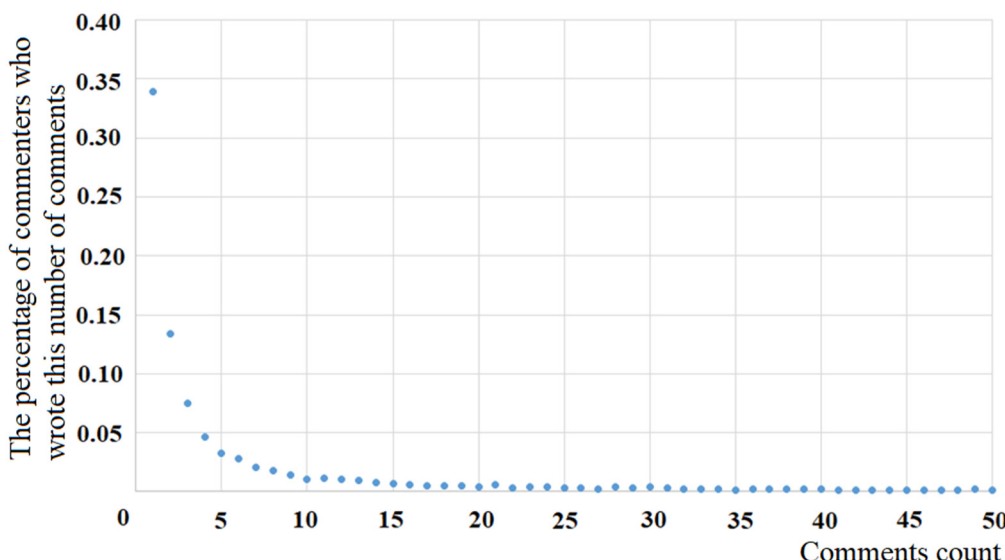

**Figure 1.** Density of distribution of commentators by their number of comments for the period from 1 January to 31 December 2020.

Note that Figure 1 shows only part of the data. Some users managed to write several hundred comments during the year (the maximum number of comments on one news item was 239), but their share is rather small. So, for clarity of presentation, the right part of the chart has been reduced, because it is uninformative.

*4.2. Processing of Observed Data*

When analyzing the observed data (see Figure 1), it is crucial to establish the distribution law that the observed distribution density is subject to. Otherwise, se of the data obtained is difficult in terms of predicting the behavior of the process and making recommendations for decision-making.

Considering the process of creating comments by users to be largely random (due to the different probability of occurrence of various news events and the degree of interest in them, etc.), let us consider the three most frequently observed distribution laws:

1. Gaussian distribution: $\rho(x) = e^{-\frac{x^2}{2 \cdot \sigma^2}} / \sigma\sqrt{2\pi}$
2. Exponential distribution: $\rho(x) = a \cdot e^{-\alpha x}$
3. Power distribution: $\rho(x) = \beta \cdot x^{-\gamma}$

If any of these distributions is fulfilled, then the observed data should be linearized in the appropriate coordinates with an acceptable value of the correlation coefficient (0.95–0.98):

1. For the Gaussian distribution: $ln\{\rho(x)\} = -ln\{\sigma\sqrt{2\pi}\} - \frac{1}{2 \cdot \sigma^2} \cdot x^2$
2. For exponential distribution: $ln\{\rho(x)\} = ln\{a\} - \alpha x$
3. For the power distribution: $ln\{\rho(x)\} = ln\{\beta\} - \gamma ln\{x\}$

The linearization of the observed data for various types of distribution is shown in Figures 2–4 ("1"—The area in which the "fluffy tail" is observed.).

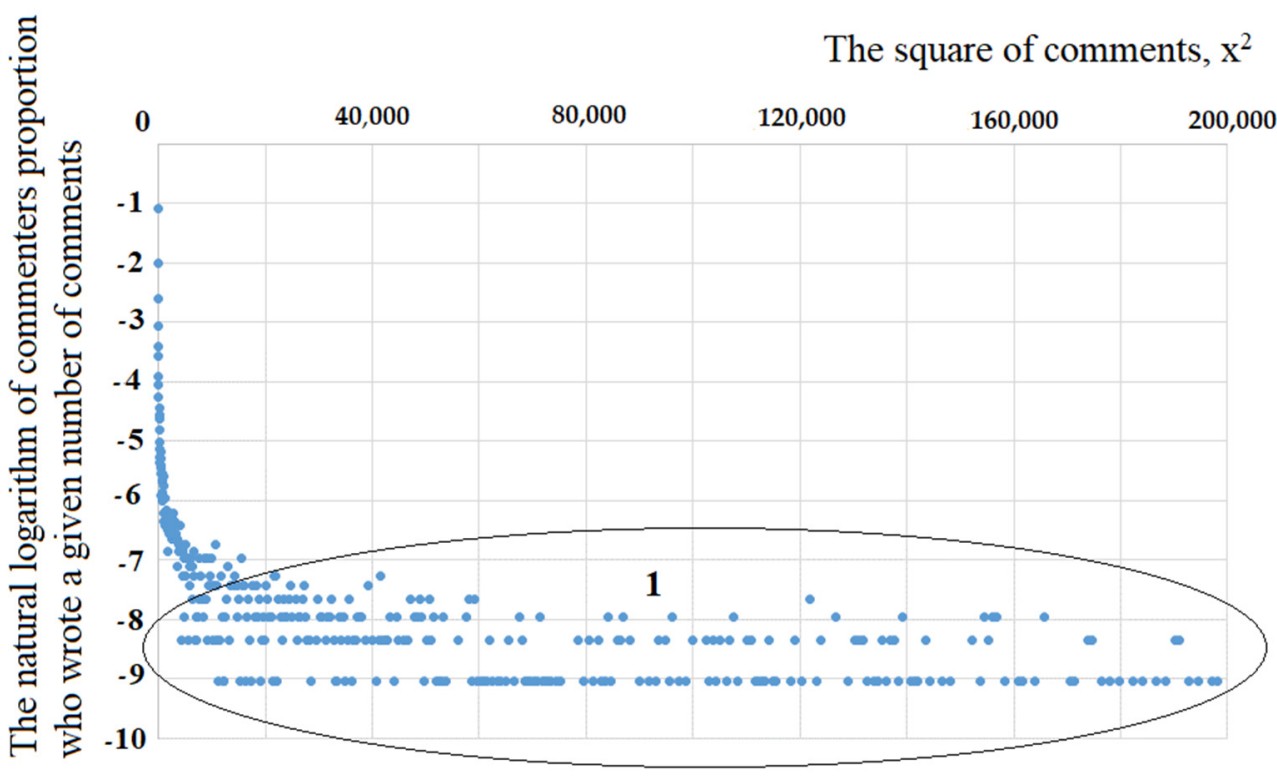

**Figure 2.** Linearization of the observed data for the Gaussian distribution.

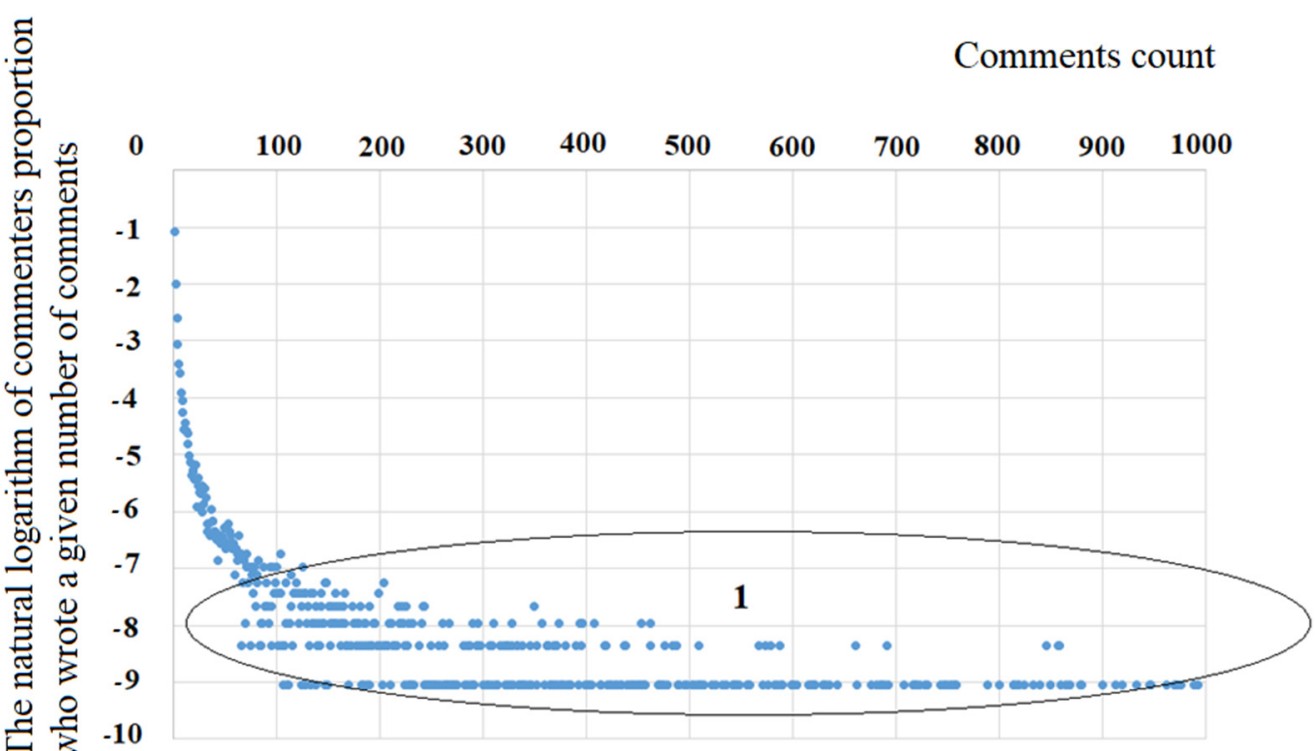

**Figure 3.** Linearization of observed data for exponential distribution.

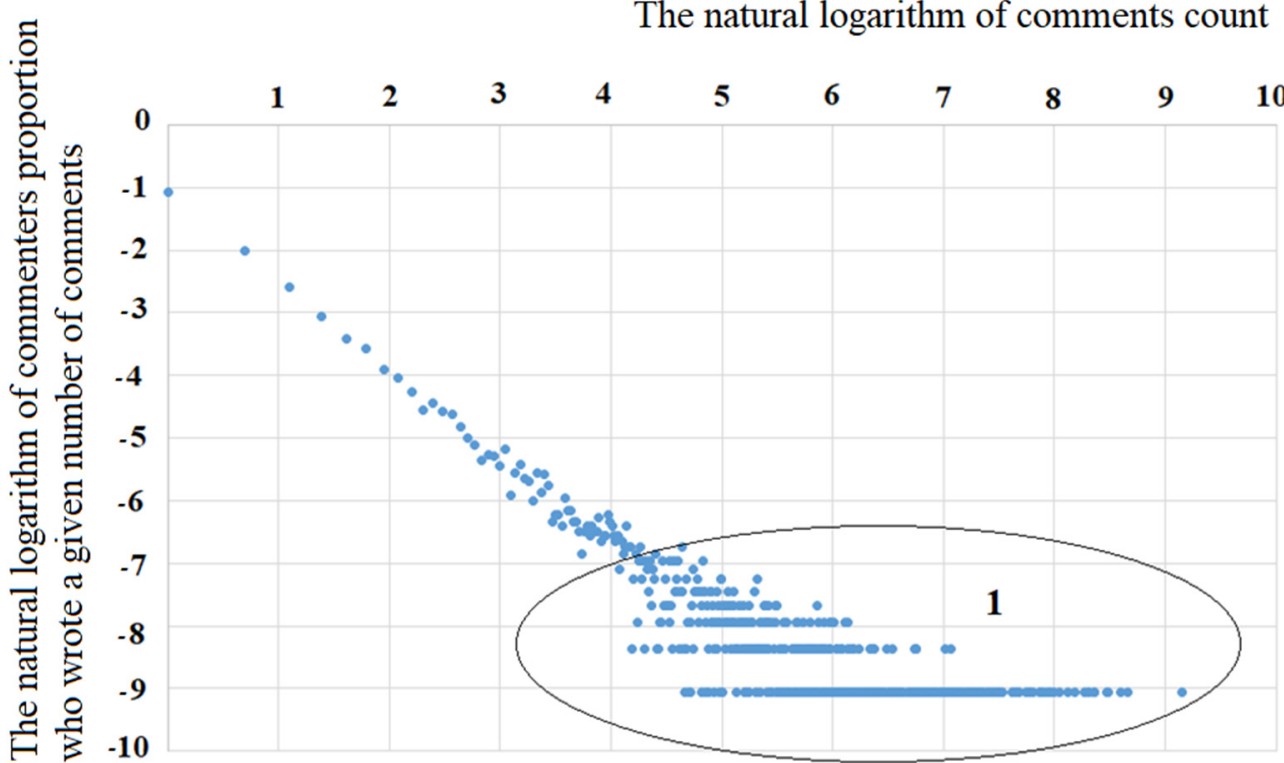

**Figure 4.** Linearization of the observed data for power distribution.

As can be seen from Figures 2–4, the best linearization is observed for approximating the observed data by the power law of distribution (see Figure 4). However, the areas shown in Figures 2–4 by the oval figure, which we called the "fluffy tail", deserve special discussion. Their appearance is due to the fact that, in addition to the so-called conscientious users, there are chatbots and users who write comments on a professional basis among the commentators. A rule can be introduced according to which unscrupulous users and chatbots can include those commentators who make more than 6–10 comments per day, as well as those who create several comments in a very short time interval (high-frequency commenting).

After appropriate purification, data can be obtained, the linearization of which, for the power law, is shown in Figure 5. There is no acceptable linearization for the exponential distribution and the Gaussian distribution. The straight line in Figure 5 shows that the trend line is well described by the linear approximation $y = -1.49 - 1.23z$, where $y = ln\{\rho(x)\}$, $z = ln\{x\}$, $ln\{\beta\} = -1.47$, and the correlation coefficient is 0.98.

In addition, to confirm the conclusion regarding linear approximation, it is possible to investigate the behavior of the residuals, and test the hypothesis that they are normally distributed with an average value equal to zero and have a homogeneous variance. The calculation of the residuals can be carried out on the basis of the actually observed values of the natural logarithm of the proportion of commentators who gave a given number of comments and the equation we obtained, for a given logarithm of the number of comments. The calculated value of the mathematical expectation for the distribution of residues is 0.25 and the variance is 0.13. The asymmetry is 0.64; the kurtosis is 0.14. Testing the slope hypothesis (two-sample F-test for variances) shows that the variance of the residuals (calculated relative to the trend line) is significantly less than the variance of the deviation of linear regression points from the average value of the observed data ($\Sigma y_i/n = \Sigma ln\{\rho(x_i)\}/n$). This is equal to 2.11 (0.13 << 2.11). Thus, the resulting regression is significant. The asymmetry characterizes the "skewness" of the distribution function, and for symmetric functions (for example, the normal distribution) it is zero (in our case, it is small and close to zero). The kurtosis characterizes the "tail" of the distribution. With large positive values for

the kurtosis, the distribution function decreases more slowly with further distance from the average value than with small ones. If the excess value is greater than zero, the distribution density graph will lie above the normal distribution graph and, for less than zero, below the graph (in our case, this is small and very close to zero). Thus, from the data obtained, it can be concluded that the distribution of residuals is very close to normal, which confirms the conclusion that the natural logarithm of the proportion of commentators who wrote these comments linearly depends on the natural logarithm of the number of comments, which confirms the fulfillment of the power law.

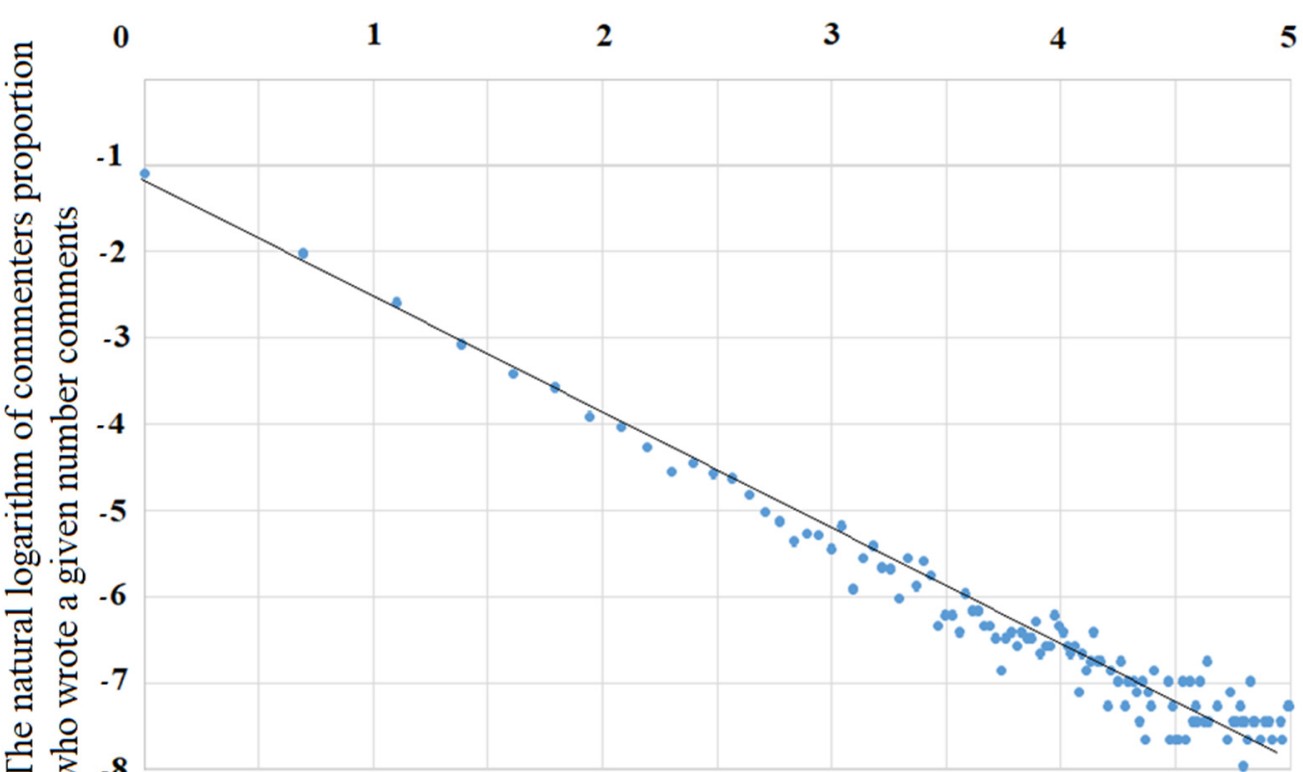

**Figure 5.** Linearization of the observed data for power distribution after cleaning unscrupulous users.

Thus, it can be assumed with great certainty that the density of the distribution of commentators by their number of comments obeys a power law.

It seems interesting to consider the dynamics of the changes in the number of comments on news of great public interest (during viewing, such types of newsfeed or blogs gain hundreds of comments) over time.

As an illustrative example, the news that appeared on the Echo of Moscow portal (https://echo.msk.ru/news/2626290-echo.html, accessed on 21 November 2021) can be chosen. On 21 November 2021: "The Public Council under the Ministry of Defense made a proposal to rename the Prague metro station in honor of Marshal Konev." The total number of comments was 221. Figure 6 shows the dynamics of changes in the number of comments on this news item over time. The number of comments at the first level (comments one news itself) was 107, at the second level (comments of the comments at the first level) 26, at the third 24, and at the fourth and more, the average time for second-level comments to appear is about 130 min.

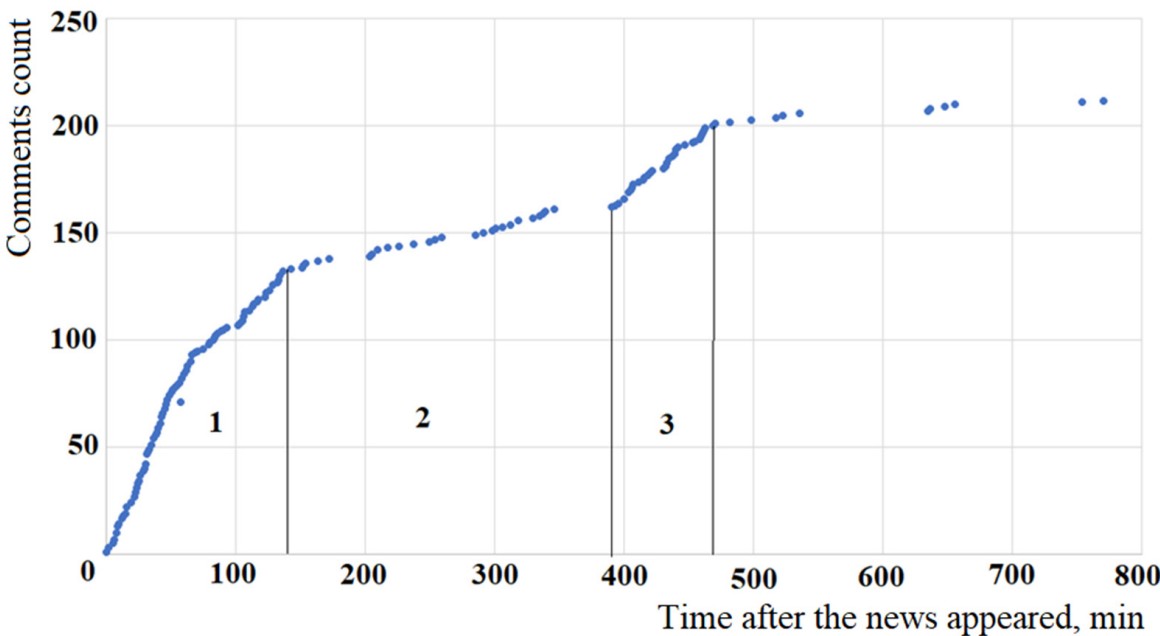

**Figure 6.** The observed dynamics of change over time, the number of comments on a news item of public interest that appeared on the portal https://echo.msk.ru/news/2626290-echo.html on 16 April 2020.

As another illustrative example, the news that appeared on the Echo of Moscow portal (https://echo.msk.ru/news/2740844-echo.html, accessed on 21 November 2021) on 12 November 2020 can be chosen: "Lavrov said that the Russian Federation has reason to believe that Navalny was poisoned on a plane or in Germany." The total number of comments was 220. The dynamics of the change in the number of comments on this news item over time is shown in Figure 7.

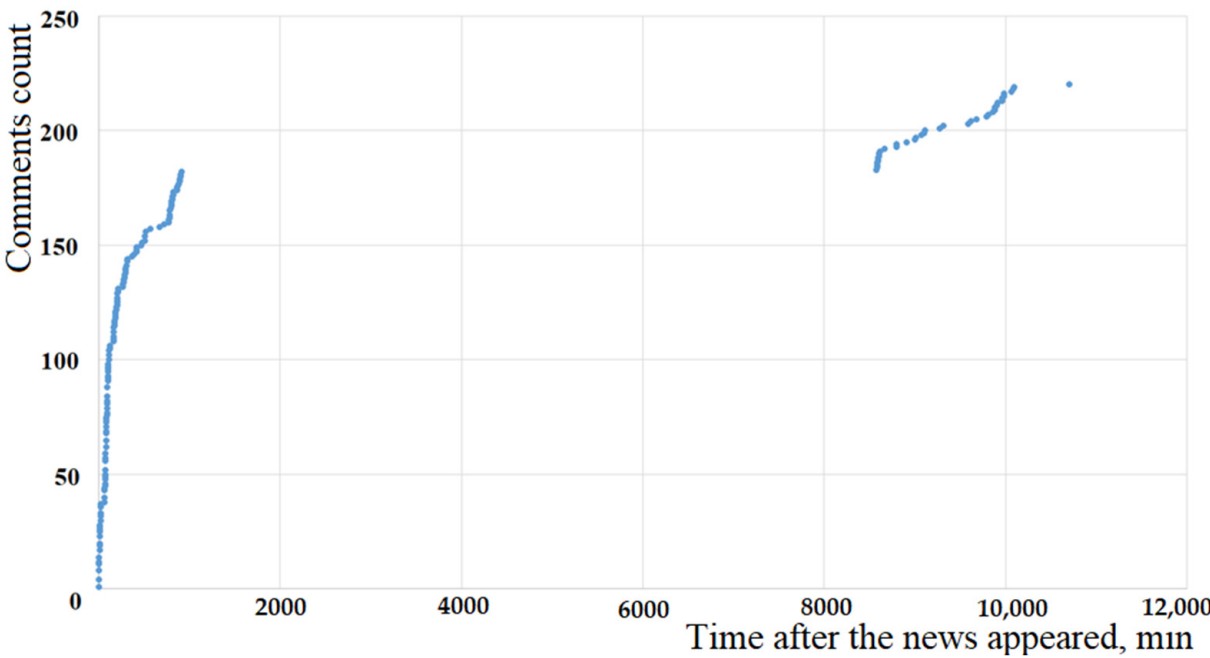

**Figure 7.** The observed dynamics of change over time, the number of comments on a news item of public interest that appeared on the portal https://echo.msk.ru/news/2740844-echo.html on 12 November 2020.

After removing the time gaps, the dynamics take the form shown in Figure 8.

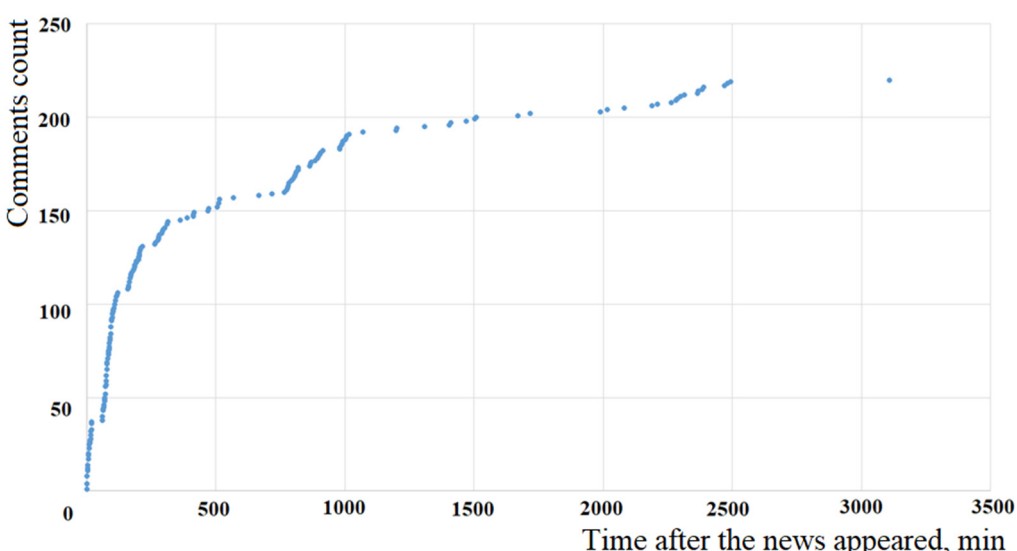

**Figure 8.** Dynamics of changes over time, the number of comments on a news item of public interest that appeared on the portal https://echo.msk.ru/news/2740844-echo.html on 12 November 2020, after removing the time gaps

The number of comments at the first level (comments on the news itself) was 93, at the second (comments on comments at the first level) 32, at the third 22, and at the fourth and more 73. The average time for second-level comments to appear is about 100 min.

It should be noted that in addition to the two-stage curves (see Figures 6 and 8), in some cases there is an S dynamic for changes in the number of comments (see Figure 9 (without removing the gaps) and Figure 10 (after removing the time gaps) for the news item "Putin has nominated Mishustin for the post of prime Minister" published on the portal https://echo.msk.ru/news/2571431-echo.html (accessed on 21 November 2021) on 15 January 2020 which received 208 comments).

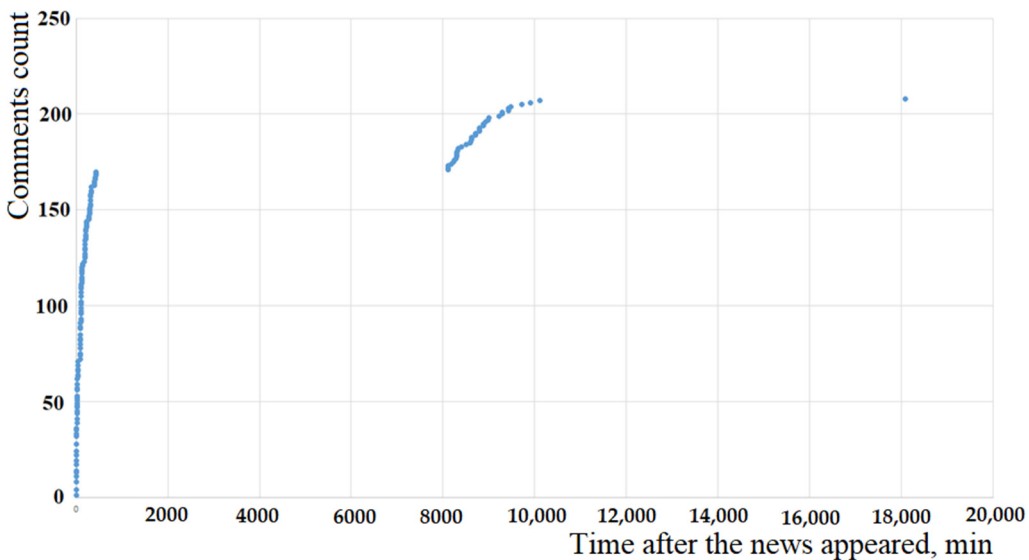

**Figure 9.** The observed dynamics of change over time, the number of comments on a news item of public interest that appeared on the portal https://echo.msk.ru/news/2571431-echo.html on 15 January 2020.

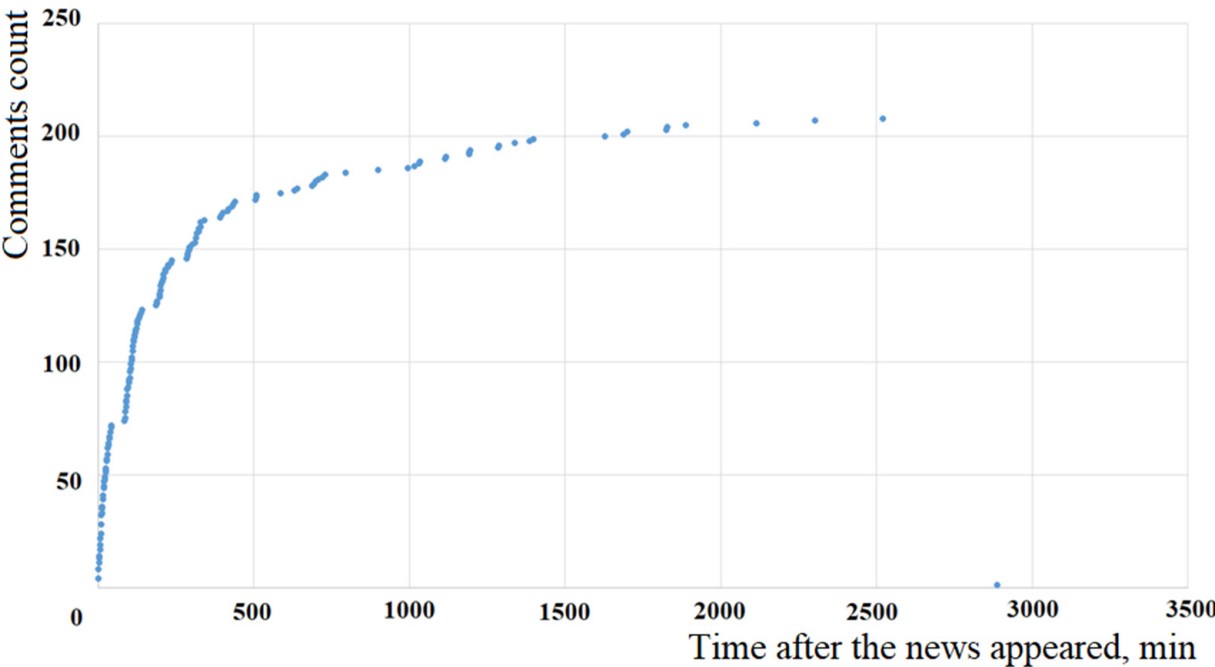

**Figure 10.** Dynamics of changes over time, the number of comments on a news item of public interest that appeared on the portal https://echo.msk.ru/news/2571431-echo.html on 15 January 2020, after removing the time gaps.

The number of comments at the first level (on the news itself) was 90, at the second (comments on comments at the first level) 38, at the third 14, and at the fourth and more 66. The average time for second-level comments to appear is about 56 min.

Note that the length of the sections of curves 1, 2 and 3 in Figures 6, 8 and 10 may be different, as well as the growth areas of the S-shaped curves.

The dynamics of the appearance of comments for news items: (1) On 16 April 2020: "The Public Council under the Ministry of Defense made a proposal to rename the Prazhskaya metro station in honor of Marshal Konev"; and (2) On 12 November 2020: "Lavrov stated that the Russian Federation has reason to believe that Navalny was poisoned on an airplane or in Germany", have a two-stage character (see Figures 6 and 8). For the news: (3) 15 January 2020: "Putin nominated Mishustin for the post of prime Minister", this is S-shaped (see Figure 10). In our opinion, this may be due to a significant difference in the average time of appearance of second-level comments (the time interval between the appearance of the first-level comment and the comment om this comment). If for the first news and for the second this is about 130 and 100 min, respectively, then for the third it is about 56 min. It should also be noted that the two-stage nature of the dynamics of commenting on the first news item is more evident than for the second, and at the same time the span until the appearance of secondary comments on the second news item is longer. For the other parameters (the total number of comments, the number of comments at the first, second and third levels), the three selected news items are close in quantitative terms.

For further study, the following theoretical research task can be formulated: what is the nature of the processes of commenting on news items and blogs, and what features of these complex social systems lead to the fact that, for the correspondence of the probability density of the distribution of comments by their number, a power law is applied and the dynamics have a complex two-stage character in many cases?

### 5. Derivation of the Power Law of the Distribution of Comments from the Stationary Fokker-Planck Equation

The Fokker-Planck equation is widely used for the analysis and modeling of non-stationary processes observed in various complex systems and allows the achievement of good agreement with the predicted behavior and observed data. Therefore, as a testable hypothesis, we assume that the Fokker-Planck equation can be used to analyze and model the appearance of comments on newsfeed and blogs.

In general, the Fokker-Planck equation has the form:

$$\frac{\partial \rho(x,t)}{\partial t} = -\frac{\partial}{\partial x}[\mu(x)\cdot\rho(x,t)] + \frac{1}{2}\frac{\partial^2}{\partial x^2}[D(x)\cdot\rho(x,t)] \tag{1}$$

where $\rho(x,t)$ is the time-dependent probability density of the distribution over states $x$ (in our case, state $x$ is the number of comments observed at time $t$), $D(x)$ is a state–dependent coefficient $x$ that determines a random change in state $x$, and $\mu(x)$ is a state-dependent coefficient $x$ that determines a purposeful change in state $x$.

In relation to our model, $D(x)$ can be interpreted as user actions caused by a spontaneous impulse that arose when reading the news or comments on it from other users, when the event described in the newsfeed or blog is not essential, but the user is willing to spend time commenting or responding to another commentator (the user has a spontaneous desire to respond to this news). $\mu(x)$ can be interpreted as purposeful actions caused by the desire to respond to a newsfeed or blog that is essential to the user, as well as to comment on another user's comment if this touches on a topic that is important from the point of view of this user (the user is constantly interested in this topic).

When analyzing the observed data, at first step we will not consider the dynamics of the appearance of comments over time, but take a static picture formed over a certain period of time (when the changes stop), so we can proceed to the stationary Fokker-Planck equation, which has the form:

$$-\frac{d}{dx}[\mu(x)\cdot\rho(x)] + \frac{1}{2}\frac{d^2}{dx^2}[D(x)\cdot\rho(x)] = 0 \tag{2}$$

Calculate the derivatives in Equation (2):

$$-\frac{d}{dx}[\mu(x)\cdot\rho(x)] = -\left[\mu(x)\cdot\frac{d\rho(x)}{dx} + \rho(x)\cdot\frac{d\mu(x)}{dx}\right]$$

$$\frac{d^2}{dx^2}[D(x)\cdot\rho(x)] = \frac{d}{dx}\left[\frac{d}{dx}[D(x)\cdot\rho(x)]\right] = \frac{d}{dx}\left[D(x)\cdot\frac{d\rho(x)}{dx} + \rho(x)\cdot\frac{dD(x)}{dx}\right] =$$

$$= D(x)\cdot\frac{d^2\rho(x)}{dx^2} + 2\frac{dD(x)}{dx}\cdot\frac{d\rho(x)}{dx} + \rho(x)\cdot\frac{d^2D(x)}{dx^2}$$

After substituting the derivatives into Equation (2), we obtain:

$$-\mu(x)\cdot\frac{d\rho(x)}{dx} - \frac{d\mu(x)}{dx}\cdot\rho(x) + \frac{1}{2}D(x)\cdot\frac{d^2\rho(x)}{dx^2} + \frac{dD(x)}{dx}\cdot\frac{d\rho(x)}{dx} + \frac{d^2D(x)}{dx^2}\cdot\rho(x) = 0 \tag{3}$$

Further, to build the model, it is necessary to make assumptions about the dependence of $D(x)$ and $\mu(x)$ on the state of $x$ and consider two conditions. Firstly, we consider the magnitude of the terms included in Equation (3), and secondly, we can assume that with the growth of state $x$ (the increase in the number of possible comments (the significance of a newsfeed or blog), the values $D(x)$ and $\mu(x)$ should also increase). Logic suggests that all terms of Equation (3) should have the same magnitude, which has $p(x)$. Both the first and the second condition will be met if the dependencies $D(x)$ and $\mu(x)$ on the state $x$ have the form: $\mu(x) = \mu_0\cdot x$ and $D(x) = D_0\cdot x^2$. In this form, the growth of $D(x)$ and $\mu(x)$

will be ensured with an increase in the state of x, and on the other hand the condition of preserving the magnitude is fulfilled. Substituting $D(x)$ and $\mu(x)$ into Equation (3) gives:

$$-\mu_0 \cdot x \cdot \frac{d\rho(x)}{dx} - \mu_0 \cdot \rho(x) + \frac{1}{2}D_0 \cdot x^2 \cdot \frac{d^2\rho(x)}{dx^2} + 2D_0 \cdot x \cdot \frac{d\rho(x)}{dx} + D_0 \cdot \rho(x) = 0$$

$$x^2 \cdot \frac{d^2\rho(x)}{dx^2} + 2\left[2 - \frac{\mu_0}{D_0}\right] \cdot x \cdot \frac{d\rho(x)}{dx} + 2\left[1 - \frac{\mu_0}{D_0}\right]\rho(x) = 0 \tag{4}$$

Denote $2\left[1 - \frac{\mu_0}{D_0}\right] = \gamma$, then:

$$x^2 \cdot \frac{d^2\rho(x)}{dx^2} + [2 + \gamma] \cdot x \cdot \frac{d\rho(x)}{dx} + \gamma \cdot \rho(x) = 0 \tag{5}$$

Equation (5) refers to equations of the Euler equation type and its solution can be found in the form: $\rho(x) = \sum_k C_k x^q$, where $C_k$ are constant coefficients at the corresponding roots of the characteristic equation, which has the form:

$$q(q-1) + [2 + \gamma]q + \gamma = 0$$

This equation has two roots: $q_1 = -1$ and $q_2 = -\gamma$. Thus, for $\rho(x)$ we obtain:

$$\rho(x) = C_1 x^{-1} + C_2 x^{-\gamma} \tag{6}$$

We find the constant coefficients $C_1$ and $C_2$ using the normalization condition of the function $\rho(x)$

$$\int_1^\infty \rho(x)dx = C_1 ln(x)\Big|_1^\infty + C_2 \frac{x^{1-\gamma}}{1-\gamma}\Big|_1^\infty \equiv 1 \tag{7}$$

Integral (7) is calculated from 1 to $\infty$, because there may be users who have made a very large number of comments to the news, but there cannot be commentators who have written less than one comment. Given that for $x \to \infty$ $ln(x)|_\infty = \infty$, then $C_1 = 0$ and, respectively, $C_2 = \gamma - 1$. Finally, we get: $\rho(x) = [\gamma - 1]x^{-\gamma}$.

Let us compare the obtained theoretical result with the observed data (see Figure 5). Linear approximation of the data presented in Figure 5 allowed us to obtain the equation: $y = -1.49 - 1.23z$, which must be compared with the equation:

$$ln\{\rho(x)\} = ln\{\gamma - 1\} - \gamma \cdot ln(x) \tag{8}$$

If $\gamma = 1.23$, then $ln(\gamma - 1) = -1.47$, which shows a very good correspondence between the theory and the observed data.

The results obtained show that, with a linear dependence of $\mu(x)$ on the state of x and a quadratic dependence of $D(x)$ on the state of $x$, the power law of dependence is the probability density of the distribution of comments by their number (states of $x$). This can be obtained from the solution of the stationary Fokker-Planck equation, and the observed data and theoretical calculations have good agreement with each other.

Special attention should be paid to this result. Its importance lies in the fact that the effects of memory and self-organization play an important role in the dynamics of social processes. However, in this case it turns out that from the Fokker-Planck equation (describing the dynamics as a whole at the macro level), the derivation of which considers a completely stochastic Markov approximation, it is possible to obtain theoretical results that are in good agreement with the observed data. We can make an assumption that the multi-directionality of a multitude of local actions and processes, each of which has both memory and self-organization, leads in the total result to the fact that memory can largely disappear as a result of the multi-direction of the ongoing micro-processes.

### 6. A Model of the Nonlinear Dynamics of the Appearance of Comments Based on the Fokker-Planck Equation

Since the use of the Fokker-Planck equation and the approach described above allow us to obtain the power law of distribution observed in practice, it is advisable to use this equation to describe the dynamics of the observed processes.

Using the method of Laplace transformations for Equation (1), it is possible to obtain (see Appendix A) the following expression for the distribution function:

$$\rho(x,t) = \int \frac{\left[\frac{[ln(x)]^2}{D_0 t} + \left[\frac{1}{2} - \frac{\mu_0}{D_0}\right] ln(x) - 1\right]}{\sqrt{2\pi D_0 t^3}} e^{-\left[\frac{[ln(x)]^2}{2D_0 t} + \left[\frac{3}{2} - \frac{\mu_0}{D_0}\right] ln(x) + \left[\frac{1}{2} - \frac{\mu_0}{D_0}\right]^2 \frac{D_0 t}{2}\right]} dt \quad (9)$$

The probability that the number of comments by the time it reaches a certain number *L* can be found by the formula (10):

$$P(L,t) = 1 - \int_0^L \left[ \int \frac{\left[\frac{[ln(x)]^2}{D_0 t} + \left[\frac{1}{2} - \frac{\mu_0}{D_0}\right] ln(x) - 1\right]}{\sqrt{2\pi D_0 t^3}} e^{-\left[\frac{[ln(x)]^2}{2D_0 t} + \left[\frac{3}{2} - \frac{\mu_0}{D_0}\right] ln(x) + \left[\frac{1}{2} - \frac{\mu_0}{D_0}\right]^2 \frac{D_0 t}{2}\right]} dt \right] dx \quad (10)$$

$$\int_0^L \left[ \int \frac{\left[\frac{[ln(x)]^2}{D_0 t} + \left[\frac{1}{2} - \frac{\mu_0}{D_0}\right] ln(x) - 1\right]}{\sqrt{2\pi D_0 t^3}} e^{-\left[\frac{[ln(x)]^2}{2D_0 t} + \left[\frac{3}{2} - \frac{\mu_0}{D_0}\right] ln(x) + \left[\frac{1}{2} - \frac{\mu_0}{D_0}\right]^2 \frac{D_0 t}{2}\right]} dt \right] dx$$

This determines the probability that the threshold *L* (for example, the maximum possible value of the number of comments) will not be reached by time *t*. The dependence of the number of comments $N(t)$ on time t will be described by the equation: $N(t) = P(L,t) \cdot L$.

We will conduct simulation modeling and analyze the theoretical results obtained. As an example, we choose *L* = 100 and three sets of values of $\mu_0$ and $D_0$ ($\mu_0$ = 0.45 и $D_0$ = 0.50 conventional units ($\mu_0 < D_0$ see curve 1 in Figure 11); $\mu_0$ = 0.50 и $D_0$ = 0.50 and $\mu_0 = D_0$ conventional units ($\mu_0 > D_0$ see curve 2 in Figure 11) and $\mu_0$ = 0.55 and $D_0$ = 0.50 conventional units ($\mu_0 > D_0$ see curve 3 in Figure 11)). Figure 11 shows the results of modeling the dynamics of changes over time in the number of comments $N(t)$ at the selected values of the model parameters $\mu_0$, $D_0$ and *L*.

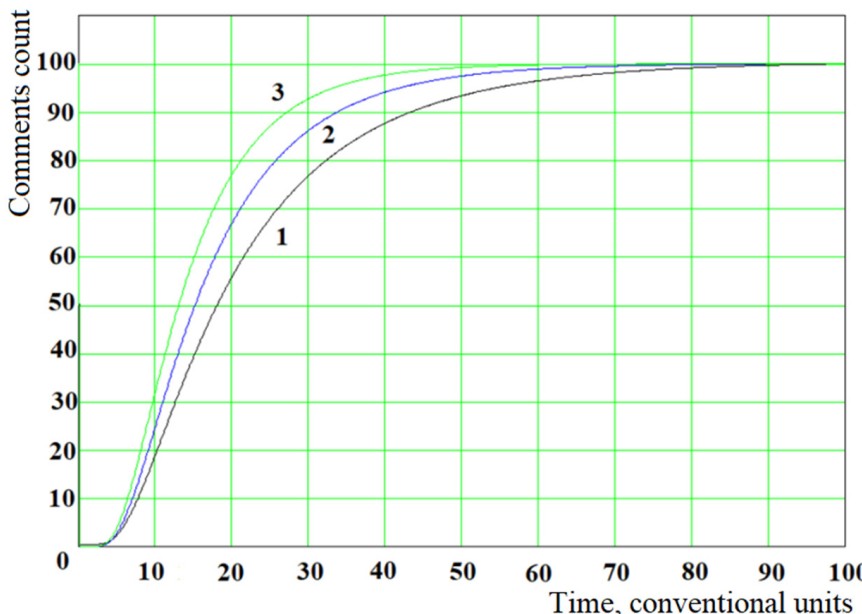

**Figure 11.** Dynamics of changes over time in the number of comments to the news in a simulation model based on the Fokker-Planck equation.

Theoretical calculations show that, with the growth of $\mu_0$ relative to $D_0$, the growth rate of the curve increases (see Figure 11).

It is important to note that the model based on the Fokker-Planck equation for all values of the parameters $\mu_0$ and $D_0$ shows the S-shaped nature of the dynamics of changes in the number of comments to the news over time, which in many cases is not consistent with the observed data (see Figures 6 and 8).

The correspondence of the theoretical model and the observed data (see Figures 6 and 8) can be obtained if we assume that two processes with different $\mu_0$ and $D_0$ can occur simultaneously. Moreover, the sum of the partial fractions of the processes should be equal to 1, i.e., $P_{total}(L,t) = \alpha_1 \cdot P_1(L,t) + \alpha_2 \cdot P_2(L,t)$, where $\alpha_1 + \alpha_2 = 1$. At the same time, one of the processes is generated by commenting on the newsfeed or blog itself, and the second by commenting on comments. To describe this, we consider the possible time delay in commenting on comments in the model. If we enter the delay time (denote it $\tau$), then the distribution function will take the form:

$$\rho(x,t-\tau) = \int \frac{\left[\frac{[ln(x)]^2}{D_0[t-\tau]} + \left[\frac{1}{2} - \frac{\mu_0}{D_0}\right]ln(x) - 1\right]}{\sqrt{2\pi D_0[t-\tau]^3}} e^{-\left[\frac{[ln(x)]^2}{2D_0[t-\tau]} + \left[\frac{3}{2} - \frac{\mu_0}{D_0}\right]ln(x) + \left[\frac{1}{2} - \frac{\mu_0}{D_0}\right]^2 \frac{D_0[t-\tau]}{2}\right]} dt$$

As we wrote earlier, this may be due to a significant difference in the average time of appearance of second-level comments (the time interval between the appearance of a first-level comment and a comment on this comment), which may lead to the implementation of two-stage dynamics in the appearance of comments.

As an example of modeling, we will choose the following model parameters for the process of commenting on the newsfeed or blog itself: $\mu_{0,1}$ = 0.55, $D_{0,1}$ = 0.50, and for the second (commenting on comments) $\mu_{0,2}$ = 0.50, $D_{0,2}$ = 0.50, $\tau$ = 50 conventional units, $\alpha_1$ = 0.75, $\alpha_2$ = 0.25 and $L$ = 100 ($\mu_{0,1} > \mu_{0,2}$ was chosen based on the assumption that commenting on the news is a more primary process for users than commenting on comments.

Figure 12 shows the results of modeling the dynamics of changes in the number of comments $N(t)$ over time, because two processes can occur in parallel. As can be seen from the simulation results presented in Figure 12, there is a good coincidence of real data (see Figures 6 and 8) and theoretical calculations (curve 1, constructed considering the time delay $\tau$). Without considering the delay, the dynamics of the news commenting process is S-shaped (see curve 2 in Figure 12), which coincides with the observed data presented in Figure 10 and is consistent with a significant difference in the average time of appearance of second-level comments for news items 1, 2 and 3 selected as an example.

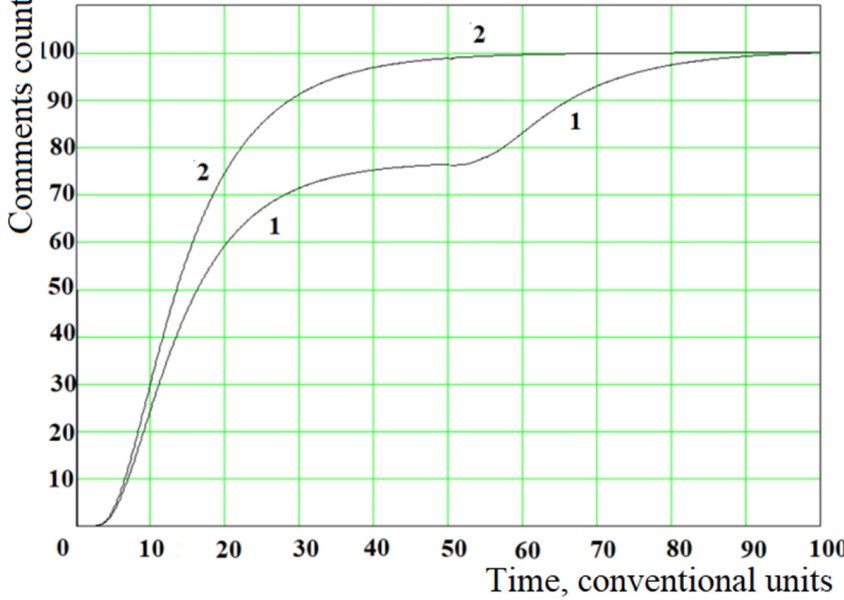

**Figure 12.** Dynamics of changes over time in the number of comments on the news in the simulation model based on the Fokker-Planck equation, considering two parallel processes.

The parallel flow of the two processes does not violate the integrity of the model, because the stationary solution of the modified Fokker-Planck equation (taking into account the delay by $\tau$) and the usual equation has the same form, which was described in the section "Derivation of the power law of the distribution of comments from the stationary Fokker-Planck equation".

## 7. Discussion

Firstly, it is possible to analyze the topics of news items that gain the largest number of comments (i.e., have the greatest public interest), make a ranking of their popularity, and study their static distributions. Further, within each group, it is possible to determine the exponent of the power law $\rho(x) = [\gamma - 1]x^{-\gamma}$. Then, considering that $\frac{\mu_0}{D_0} = 1 - \frac{\gamma}{2}$, it is possible to determine the value of $\frac{\mu_0}{D_0}$ by which it is possible to judge for which types of news and messages purposeful commenting is predominant (an increase in the ratio of $\frac{\mu_0}{D_0}$), and for which this is "random" (a decrease in the ratio of $\frac{\mu_0}{D_0}$). This will allow prediction in the future as to what news item may cause what user behavior, and how they may influence public opinion.

Secondly, using the dynamic distribution functions obtained in this work, it is possible to analyze the observed processes of commenting on newsfeed and blogs. Further, based on this, it is possible to determine the parameters of the model $\mu_0$, $D_0$ and $\tau$ for various types of news, which can also allow prediction in the future what news may cause what user behavior, and how this may influence public opinion.

In conclusion, we note that the complex nature of the dynamics of processes in complex social systems can be described, not only based on models created based on the Fokker-Planck equation. For example, in [41–48], models were developed by the authors specifically to describe the stochastic dynamics of changes in the state of complex social systems. These models take into account the processes of self-organization and memory availability. To create this model, graphical diagrams of the probabilities of transitions between possible states of the described systems were considered taking into account previous states. This method allows the taking into account memory, and describes not only Markov but also non-Markov processes. Using this approach, a nonlinear differential equation of the second order was derived, which allows the setting and solution of problems for determining the probability density function of the amplitude of deviations of parameters describing the observed processes of a non-stationary time series, depending on the values of the time interval of its determination and the depth of memory accounting. The differential equation obtained during the study contains not only terms responsible for random change (diffusion) and ordered change (destruction), but also a term that is responsible for the possibility of self-organization, which significantly distinguishes it from the Fokker-Planck equation. Within the framework of the models developed by [41–48], it is possible to describe processes whose dynamics have both an S-shaped character for changes and a two-stage process.

The novelty of our work in comparison with the works of our predecessors is that, by using a stationary version of the Fokker-Planck equation for the data observed in practice, a power law of the distribution of their parameters can be obtained that is consistent with them. In this case, it can be made a pre-position that the multidirectional nature of many local actions and processes, each of which has both memory and self-organization, leads in summary to the fact that memory can largely disappear and the process in a generalized form becomes Markovsky. This allowed us, under certain assumptions for coefficients in the Fokker-Planck equation, to obtain from its stationary form a power law of distribution for the number of comments on news and blogs. As shown in our paper, the theory aligns well with the data observed in reality.

Secondly, assuming that the Fokker-Planck equation under certain circumstances can be applied to describe the dynamics in the systems in question (for example, based on what is described above) we considered the temporal dependencies of the appearance of comments on various news and found that it can be both S-shaped in nature and have a more complex-two-staged form, which can be explained within the framework of using the Fokker-Planck equation only by the presence of two processes and delay time.

## 8. Conclusions

The results obtained in the work allow us to draw several conclusions:

1.  The stationary distribution of news observed in practice by the number of comments to on it corresponds to the power law: $\rho(x) = [\gamma - 1]x^{-\gamma}$, where $\rho(x)$ is the share of news items in their total number having $x$ comments, and $\gamma$ is the exponent.

2. The dynamics of changes over time in the number of comments to a newsfeed or blog can have both an S-shaped form and a two-stage one, which may be due to a significant difference in the average time of appearance of comments at the second level (the time interval between the appearance of a comment at the first level and a comment on this comment), i.e., the value of the average delay.

3. The power law of dependence observed in practice is the stationary probability density of the distribution of news by the number of comments (states $x$) which can be obtained from the solution of the stationary Fokker-Planck equation if some assumptions are made during its derivation. We assume that the coefficient $\mu(x)$ responsible in the Fokker-Planck equation for a purposeful change in the state of the system $x$ ($x$ is the current number of comments on the news) linearly depends on the state $x$, and the coefficient $D(x)$ responsible for a random change depends on $x$ quadratically. All this suggests that the Fokker-Planck equation can be used to describe processes in complex network structures.

4. The solution of the unsteady Fokker-Planck equation under the assumptions of the linear dependence of $\mu(x)$ on the state of $x$ and the quadratic dependence of $D(x)$ on the state of $x$ allows us to obtain an equation for the probability density of transitions between the states of the system per unit of time, which are in good agreement with the observed data, taking into account the effect of the delay time between the appearance of the first level comment and the comment on this comment.

5. The models developed based on the Fokker-Planck equation are in good agreement with the observed data, which makes it possible to create algorithms for monitoring and predicting the evolution of public opinion of users of news information resources.

**Author Contributions:** D.Z.: conceptualization, formal analysis, writing-review & editing; J.P.: methodology, visualization; V.K.: data curation, writing-original draft. All authors have read and agreed to the published version of the manuscript.

**Funding:** This research was supported by the Russian Science Foundation (RSF), grant no. 22-21-00109 "Development of the dynamics forecasting models of social moods based on the analysis of text content time series of social networks using the Fokker-Planck and nonlinear diffusion equations".

**Institutional Review Board Statement:** Not applicable.

**Informed Consent Statement:** Not applicable.

**Data Availability Statement:** Not applicable.

**Conflicts of Interest:** The authors declare no conflict of interest.

**Appendix A**

One of the solutions of the Fokker-Planck equation can be obtained as follows. Using the method of Laplace transformations for Equation (1), we can write:

$$s\overline{G(s,x)} - \rho(0,x) = -\frac{d}{dx}\left[\mu(x)\cdot\overline{G(s,x)}\right] + \frac{1}{2}\frac{d^2}{dx^2}\left[D(x)\cdot\overline{G(s,x)}\right] \tag{A1}$$

Considering that at time $t = 0$ (the beginning of the process) there are no comments, then: $\rho(0,x) = 0$.

Further, substituting into Equation (A1) the corresponding derivatives and dependencies $\mu(x)$ and $D(x)$ (the choice of which was discussed earlier, and their use leading to the results of the distribution of the number of comments according to the power law observed in reality), we obtain:

$$x^2\cdot\frac{d^2\overline{G(s,x)}}{dx^2} + 2\left[2 - \frac{\mu_0}{D_0}\right]\cdot x\cdot\frac{d\overline{G(s,x)}}{dx} + 2\left[1 - \frac{\mu_0 + s}{D_0}\right]\overline{G(s,x)} = 0 \tag{A2}$$

We are looking for a solution to this equation in the form: $\overline{G(s,x)} = \sum_k C_k x^q$, where $C_k$ are the coefficients for the roots of the characteristic equation, which has the form: $q(q-1) + 2\left[2 - \frac{\mu_0}{D_0}\right]q + 2\left[1 - \frac{\mu_0 + s}{D_0}\right] = 0$. Let us finds the roots of the characteristic equation.

$$q_{1,2} = -\frac{\left[3 - 2\frac{\mu_0}{D_0}\right]}{2} \pm \frac{\sqrt{\left[1 - 2\frac{\mu_0}{D_0}\right]^2 + 8\frac{s}{D_0}}}{2}$$

We write it down as follows:

$$\overline{G(s,x)} = x^{-\frac{[3-2\frac{\mu_0}{D_0}]}{2}}\left\{C_1 x^{\frac{\sqrt{[1-2\frac{\mu_0}{D_0}]^2+8\frac{s}{D_0}}}{2}} + C_2 x^{-\frac{\sqrt{[1-2\frac{\mu_0}{D_0}]^2+8\frac{s}{D_0}}}{2}}\right\}$$

Given that $\gamma = 2\left[1 - \frac{\mu_0}{D_0}\right]$ we write:

$$\overline{G(s,x)} = C_1 \cdot x^{-\frac{[\gamma+1]-\sqrt{[\gamma-1]^2+8\frac{s}{D_0}}}{2}} + C_2 \cdot x^{-\frac{[\gamma+1]+\sqrt{[\gamma-1]^2+8\frac{s}{D_0}}}{2}}$$

For $s \to \infty$ $(t \to 0)$ $\rho(x,0)$ for any $x$ must be equal to 0, so $C_1$ should be put equal to 0 $\frac{-[\gamma+1]+\sqrt{[\gamma-1]^2+8\frac{s}{D_0}}}{2} \to +\infty$ and $x \to +\infty$). Using the normalization condition (for the image, the integral from 1 to $\infty$ must be equal to $\frac{1}{s}$), we find the coefficient with $C_2$:

$$\frac{2C_2}{-[\gamma+1]-\sqrt{[\gamma-1]^2+8\frac{s}{D_0}}+2}\cdot x^{-\frac{[\gamma+1]+\sqrt{[\gamma-1]^2+8\frac{s}{D_0}}}{2}+1}\Big|_1^{\infty} = \frac{1}{s}$$

$$C_2 = \frac{[\gamma-1]+\sqrt{[\gamma-1]^2+8\frac{s}{D_0}}}{2s}$$

$$\overline{G(s,x)} = \frac{[\gamma-1]+\sqrt{[\gamma-1]^2+8\frac{s}{D_0}}}{2s}\cdot x^{-\frac{[\gamma+1]+\sqrt{[\gamma-1]^2+8\frac{s}{D_0}}}{2}}$$

Substitute $\gamma$ and get:

$$\alpha = \frac{1-2\frac{\mu_0}{D_0}}{2} = \frac{1}{2} - \frac{\mu_0}{D_0}$$

$$\frac{3-2\frac{\mu_0}{D_0}}{2} = 1 + \frac{1-2\frac{\mu_0}{D_0}}{2} = 1 + \alpha$$

$$\beta = \frac{1}{2}\sqrt{\frac{8}{D_0}} = \sqrt{\frac{2}{D_0}}$$

$$k = \frac{D_0}{8}\left[1-2\frac{\mu_0}{D_0}\right]^2 = \frac{D_0}{2}\left[\frac{1}{2}-\frac{\mu_0}{D_0}\right]^2 = \left[\frac{\alpha}{\beta}\right]^2$$

$$x^{-\beta\sqrt{k+s}} = e^{-[\beta\cdot ln(x)]\sqrt{k+s}}$$

Let us writes this:

$$\overline{G(s,x)} = \left[\alpha\cdot\frac{e^{-[\beta\cdot ln(x)]\cdot\sqrt{k+s}}}{s} + \beta\cdot\frac{\sqrt{k+s}\cdot e^{-[\beta\cdot ln(x)]\cdot\sqrt{k+s}}}{s}\right]\cdot x^{-[1+\alpha]}$$

We find the original $\frac{e^{-[\beta\cdot ln(x)]\cdot\sqrt{k+s}}}{s}$ and the original $\beta\cdot\frac{\sqrt{k+s}\cdot e^{-[\beta\cdot ln(x)]\cdot\sqrt{k+s}}}{s}$ we find by differentiating the original $\frac{e^{-[\beta\cdot ln(x)]\cdot\sqrt{k+s}}}{s}$ by $ln(x)$.

$$\frac{d}{d(ln(x))}\left[\frac{e^{-[\beta\cdot ln(x)]\cdot\sqrt{k+s}}}{s}\right] = -\beta\cdot\frac{\sqrt{k+s}\cdot e^{-[\beta\cdot ln(x)]\cdot\sqrt{k+s}}}{s}$$

$$\overline{G(s,x)} = \left[\alpha\cdot\frac{e^{-[\beta\cdot ln(x)]\cdot\sqrt{k+s}}}{s} - \frac{d}{d(ln(x))}\left[\frac{e^{-[\beta\cdot ln(x)]\cdot\sqrt{k+s}}}{s}\right]\right]\cdot x^{-[1+\alpha]}$$

$\frac{e^{-[\beta\cdot ln(x)]\cdot\sqrt{k+s}}}{s} = \frac{1}{s}\cdot e^{-y\cdot\sqrt{k+s}}$, where $[\beta\cdot ln(x)] = y$.

Dividing an image by $s$ is analogous to integrating over t of the original $e^{-y\cdot\sqrt{k+s}}$. Let us find this original:

$$e^{-[\beta\cdot ln(x)]\cdot\sqrt{k+s}} \fallingdotseq \frac{\beta\cdot ln(x)}{2\sqrt{\pi t^3}}\cdot e^{-\frac{[\beta\cdot ln(x)]^2}{4t}}\cdot e^{-kt}$$

$$\frac{e^{-[\beta\cdot ln(x)]\cdot\sqrt{k+s}}}{s} \fallingdotseq \int \frac{\beta\cdot ln(x)}{2\sqrt{\pi t^3}}\cdot e^{-\frac{[\beta\cdot ln(x)]^2}{4t}}\cdot e^{-kt}dt$$

$$\frac{d}{d(ln(x))}\left[\frac{e^{-[\beta \cdot ln(x)]\cdot\sqrt{k+s}}}{s}\right] \risingdotseq \int \frac{\beta \cdot ln(x)}{2\sqrt{\pi t^3}}\left[1 - \frac{\beta^2}{2t}\cdot ln(x)\right]\cdot e^{-\frac{[\beta \cdot ln(x)]^2}{4t}}\cdot e^{-kt}dt$$

After making all the necessary substitutions, we get the following expression for the distribution function:

$$\rho(x,t) = \int \frac{\left[\frac{[ln(x)]^2}{D_0 t} + \left[\frac{1}{2} - \frac{\mu_0}{D_0}\right]ln(x) - 1\right]}{\sqrt{2\pi D_0 t^3}}e^{-\left[\frac{[ln(x)]^2}{2D_0 t} + \left[\frac{3}{2} - \frac{\mu_0}{D_0}\right]ln(x) + \left[\frac{1}{2} - \frac{\mu_0}{D_0}\right]^2\frac{D_0 t}{2}\right]}dt \qquad (A3)$$

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
