# Peer review of "Description of the Distribution Law and Non-Linear Dynamics of Growth of Comments Number in News and Blogs Based on the Fokker-Planck Equation"

_mathematics, doi:10.3390/math10060989_

Round 1

Reviewer 1 Report

Referee Report on

Mathematical sociology of news feeds: derivation of the power law of the distribution of the number of comments from the Fokker - Planck equation and a model of the nonlinear dynamics of their change

This paper is well prepared, and it introduces a novel study about the power law of the distribution of the number of comments from the Fokker - Planck equation.

I recommend to accept this paper after minor revisions:

  1. In page 16, line 596 and in page 17, line 610 and line 626, there are non English letters.
  2. In page 11, line 447: the authors mention about the the correlation coefficient to verify that the linear models (in the previous page after taking the natural logarithm for the nonlinear models) are appropriate, but this is not enough.

        To verify the regression model is adequate, they have to do the following:

  1. Test the Hypothesis regarding the slope coefficient.
  2. Verify that the Residuals are normally distributed with mean 0.
  3. Verify that the residuals have homogeneous variance.

Reviewer 2 Report

Please see the comments in the attached file

Reviewer 3 Report

The paper is on the stationary and dynamic distributions of news based on the number of comments. The processing of the observed data obtained from the information portal demonstrates that static distribution of news by the number of comments to that news obeys a power law. The paper accordingly shows that the power law of observed dependence in practice is the stationary probability density of the news distribution by the number of comments (states x). The solution of the unsteady Fokker–Planck differential equation with the assumptions enabled the obtaining of an analytical equation for the probability density of transitions between the states of the system per unit of time.

The study provides important contributions to the literature. The content and subject matter are worthy of investigation. The observation attitude of handling data is a right approach through distributions. The steps provided for the mathematical model based on Fokker–Planck differential equation are given right as well.

Please find below my humble suggestions:

The title can be reconsidered and shortened.

In addition, some of the typos and grammatical errors can be corrected after proofreading.

I recommend that language be revised like contractions can be removed like “Let’s” needs to be written “Let us”.

The structure of the abstract can be revisited and it can be written in one single paragraph.

The novelty and aim of the work can be stated in a more evident way.

The comparisons of the work with previous ones can be written to put forth the novel aspects of the study.

There are some typo errors like this one: “At First, we downloaded the news range we were interested in using a special” in that sentence, “f” should be written in lower case. Such typos can be eliminated after a proofreading process.

Besides this, the figure quality needs to be increased according to the journal writing guidelines. The captions on x and y coordinates may not be written in bold either.

Some equations have been written in bold, they can be made consistent throughout the text.

The mathematical denotations in the text and the notations in the equations need to be in the same writing format (i.e.: line 577 page 15).

Including the data part and the mathematical part, the right steps have been provided correctly. The structure of the paper is also in line with the fluency of the text, which is good for the readers. The language is also good and academically appropriate.

Both in terms of presentation and content, it is an interesting article.

If these considerations are handled, the paper can be published.

Yours faithfully,

Round 2

Reviewer 2 Report

Please see my comments in the attached file

Author Response

Dear Mr. Reviewer,

We would like to thank you for your attentive attitude to our work and a large number of useful and interesting comments regarding our approach and the results obtained. Your comments are very valuable to us and we will rely on them in our further research.

Dear Mr. Reviewer,

Our main position is that under certain assumptions on the appearance of the coefficients of "draft" and "diffusion" from the Fokker-Planck equation (which describes Markov processes that do not take into account memory), a power law of the distribution of the share of commentators by comments can be obtained, which is very well consistent with the processes observed in practice. Social processes can have a certain amount of memory. However, the superposition of a multitude of multidirectional microprocesses with memory generally leads to stochasticity, which erases the memory of individual microprocesses in the overall picture. The resulting theoretical model is in good agreement with the processes observed in practice, and you are unlikely to deny this result.

Best regards,

team of authors